# Structural Changes in Trabecular Bone, Cortical Bone and Hyaline Cartilage as Well as Disturbances in Bone Metabolism and Mineralization in an Animal Model of Secondary Osteoporosis in *Clostridium perfringens* Infection

**DOI:** 10.3390/jcm11010205

**Published:** 2021-12-30

**Authors:** Agnieszka Tomczyk-Warunek, Tomasz Blicharski, Siemowit Muszyński, Ewa Tomaszewska, Piotr Dobrowolski, Rudolf Blicharski, Jaromir Jarecki, Anna Arczewska-Włosek, Sylwester Świątkiewicz, Damian Józefiak

**Affiliations:** 1Chair and Department of Rehabilitation and Orthopaedics, Medical University in Lublin, 20-090 Lublin, Poland; a.tomczykwarunek@gmail.com (A.T.-W.); blicharskirudolf@gmail.com (R.B.); jaromirj@interia.pl (J.J.); 2Department of Biophysics, University of Life Sciences in Lublin, 20-950 Lublin, Poland; siemowit.muszynski@up.lublin.pl; 3Department of Animal Physiology, University of Life Sciences in Lublin, 20-950 Lublin, Poland; ewaRST@interia.pl; 4Department of Functional Anatomy and Cytobiology, Maria Curie-Skłodowska University, 20-033 Lublin, Poland; piotr.dobrowolski@umcs.lublin.pl; 5Department of Animal Nutrition and Feed Science, National Research Institute of Animal Production, Krakowska St. 1, 32-083 Balice, Poland; anna.arczewska@izoo.krakow.pl (A.A.-W.); s.swiatkiewicz@izoo.krakow.pl (S.Ś.); 6Department of Animal Nutrition, Faculty of Veterinary Medicine and Animal Science, Poznań University of Life Sciences, Wołyńska 33, 60-637 Poznań, Poland; damian.jozefiak@up.poznan.pl

**Keywords:** osteoporosis, metabolic bone disorders, *Clostridium perfringens*, gastrointestinal, microflora, short-term infection

## Abstract

There is no information regarding whether changes in the microbiological balance of the gastrointestinal tract as a result of an infection with *Clostridium perfringens* influence the development of metabolic bone disorders. The experiment was carried out on male broiler chickens divided into two groups: control (*n* = 10) and experimental (*n* = 10). The experimental animals were infected with *Clostridium perfringens* between 17 and 20 days of age. The animals were euthanized at 42 days of age. The structural parameters of the trabecular bone, cortical bone, and hyaline cartilage as well as the mineralization of the bone were determined. The metabolism of the skeletal system was assessed by determining the levels of bone turnover markers, hormones, and minerals in the blood serum. The results confirm that the disturbed composition of the gastrointestinal microflora has an impact on the mineralization and metabolism of bone tissue, leading to the structural changes in cortical bone, trabecular bone, and hyaline cartilage. On the basis of the obtained results, it can be concluded that changes in the microenvironment of the gastrointestinal tract by infection with *C. perfringens* may have an impact on the earlier development of osteoporosis.

## 1. Introduction

Microorganisms living on the surface of the intestinal mucosa have a crucial impact on maintaining a proper intestinal balance. Gastrointestinal tract colonization by newborn organisms occurs immediately after delivery [1]. The number of microorganisms in the digestive tract is equal to the number of cells building the human body [2]. It is the largest ecosystem formed by bacteria, but it is poorly diversified [3]. Generally, the digestive tract is inhabited by *Bacteroidetes*, *Firmicutes* (e.g., *Clostridium* and *Eubacterium*), *Proteobacteria*, *Verrucomicrobia*, *Spirochaeates*, and *Cyanobacteria*. The most dominant are two groups of saprophytic bacteria: *Bacteroidetes* and *Firmicutes*. These saprophytes play a very important role in the proper functioning of the whole organism. They provide the necessary nutrients, affecting the synthesis of certain vitamins as well as the absorption of minerals [4]. The proper balance in the microflora of the digestive system protects the body against infections as well as affects the absorption of nutrients [5]. Disorders in the intestinal microenvironment, such as, for example, the multiplication and prevalence of *Clostridium perfringens (C. perfringens)*, negatively affect the functioning of this ecosystem and contribute to the occurrence of many diseases, including those that may contribute to the disturbance of bone metabolic processes [6,7].

*C. perfringens* is a strictly anaerobic, gram-positive bacterium that forms spores, previously called *Bacillus aerogenes capsulatus*. This bacterium commonly lives in the environment and naturally occurs in the human digestive tract. However, not everyone is a carrier of *C. perfringens*; only 5 to 31% of people have this strain in their digestive tract. The number of carriers increases with age [8].

*C. perfringens* produces a toxin, and ingestion of it is one of the most common causes of food poisoning [9,10]. It is estimated that consumption of products contaminated with *C. perfringens* enterotoxins is the second most common cause of foodborne bacterial diseases in the USA, with almost a million cases per year [11]. As a result of poisoning, approximately seven people in the United States of America and 50 to 100 in the British Isles die each year [12]. The most common source of infection is meat preparations, including poultry and beef (up to 90% of cases). Due to the ban on the use of AGP (antibiotic growth promoters), the number of cases of necrotizing enteritis in poultry with the etiology of *C. perfringens* has increased, which increases the risk of human illness. This microorganism also exists in the environment of these animals, which results in the possibility of infection of obtained animal products [13].

Available literature confirms that *C. perfringens*, along with *Clostridium difficile*, is the cause of antibiotic-associated diarrhea (AAD), which is an important factor in morbidity and mortality, especially in hospitals, where up to 15% of AAD cases are caused by this bacterium [14,15,16,17,18]. Many risk factors for AAD have been described, including age, length of hospital stay, and the administration of broad-spectrum cephalosporins, broad-spectrum penicillin, and clindamycin [15]. Risk-associated procedures include tube feeding, enemas, endoscopy, and gastrointestinal surgery. Administration of anti-cancer drugs and proton pump inhibitors increase the risk of morbidity [15,18].

Research conducted in the recent years confirms that chicken (*Gallus gallus domesticus*) can be an appropriate animal model for preclinical growth studies, which is associated with their rapid development rate [19]. They reach somatic maturity at the age of 42 days, and high daily growths allow to observe the noticeable changes in bone metabolism in a short time, which makes them widely used in experiments in which the skeletal system is studied [20,21]. Research conducted in recent years shows that this animal model is also suitable for assessing the effects of malnutrition of children on the development of their skeletal system [19]. Therefore, these animals can be an appropriate model for assessing the effect of disorders in the intestinal microenvironment on skeletal development [20,22].

Disorders in gastrointestinal homeostasis have a negative effect on the skeletal system [23] by the decrease in bone mass, preceded by irregularities in the process of mineralization, which may contribute to an increased risk of osteoporosis [7]. However, the currently available literature does not provide information on the effect of *C. perfringens* infection on bone mineralization and metabolism. 

The aim of this study was to examine whether disturbances in the gastrointestinal microenvironment as a result of *C. perfringens* infection affect the homeostasis of the skeletal system and promote the development of secondary osteoporosis. For this purpose, chickens were selected as the model animal. The content of selected minerals, hormones, and markers of bone turnover in the blood serum as well as the mechanical, geometric, and structural properties of the bones were examined. The basal histomorphometry of the trabecular bone, articular and growth plate cartilage, and the collagen structure of cortical and cancellous bone was examined.

## 2. Material and Methods

### 2.1. Animal Model

The experiment was approved by the 2nd Local Ethical Commission for Animal Testing in Krakow (No. 1186/2015).

The study was carried out at the experimental facility of the National Research Institute of Animal Production, Balice, Poland. The experiment was carried out on Ross 308 male chicken broilers (*Gallus gallus domesticus*). Healthy one-day-old chicks were used in the experiment. The animals were kept in standard breeding conditions. The birds were fed ad libitum with suitable feed for different stages of development and age, with constant access to fresh water. Upon arrival, animals were randomly divided into two groups: control and experimental. There were 10 replicates of 5 birds in each group. Birds from experimental group were housed in adjacent but separated pens animal isolation facility. In order to disturb the biological balance of the intestinal microflora, birds from the experimental group were infected with *C. perfringens*. In the experimental challenge model, for 3 days, between 17th and 20th day of age, birds from the experimental group received feed with the addition of an inoculate from 3 strains of *C. perfringens* (10^6^ CFU). On day 42, one animal from each replicate was randomly taken from both experimental groups and weighed. The total of 20 animals were euthanized by cutting the carotid arteries. Blood was collected and clotted in tubes and centrifuged (3000× *g* for 15 min) to obtain serum. The serum was separated and frozen at −60 °C. Confirmation of the changes in the gastrointestinal homeostasis was assessed during the euthanasia of the animals. Standard post-mortem veterinary examination was performed to verify the presence of gastrointestinal inflammations characteristic for *C. perfringens,* like infections in the intestinal mucosa. The animals from the experimental group showed hyperemia and necrotic changes in the intestinal mucosa in the small intestine. In the entire intestine, lesions were repeatedly found. The presence of gas and dark brown fluid was observed in the lumen of the gastrointestinal tract. Immediately after euthanasia, both tibiae were isolated from all animals. The bones were cleaned of soft tissues, weighed, measured.

### 2.2. Blood Serum Analysis

Analysis of bone turnover markers, growth hormone, calcitriol, and insulin activity were determined using commercial chicken-specific ELISA (Enzyme-Linked Immunosorbent Assay) kits: growth hormone (No. E004c, EiAab Science, Wuhan, China), insulin-like growth factor 1 (IGF-1, No. E0210Ch, Bioassay Technology Laboratory, Shanghai, China), osteocalcin (No. E-EL-Ch1115 Elabscience, Wuhan, China), and osteoprotegerin (No. E0237Ch, Bioassay Technology Laboratory, Shanghai, China), calcitriol (No. 10019eiaab, EIAab Science, Wuhan, China), and insulin (No. E0448c, EiAab Science, Wuhan, China). The analyses were carried out according to the manufacturers’ protocols. The Benchmark Plus microplate reader (Bio-Rad Laboratories, Inc., Hercules, CA, USA) was used to read the results obtained.

The serum levels of total calcium, phosphorus and alkaline phosphatase were determined. The analyzes were performed by photocolorimetric method using the Mindray BS-120 biochemical analyzer (BioMedical Electronics, Shenzhen, China) and ready-made biochemical reagent kits (Alpha Diagnostics, Warsaw, Poland). The analyzes were carried out according to the manufacturer’s protocol.

### 2.3. Mechanical Properties of the Tibia

Bone strength properties were determined during the three-point bending test performed on a Zwick Z010 testing machine (Zwick GmbH & Company KG, Ulm, Germany) equipped with measuring head of an operation range of 10 kN and coupled with computer with TestXpert II 3.1 software (Zwick GmbH & Company KG, Ulm, Germany). The bones were placed horizontally on two supports, and the distance between them was 40% of the total bone length. The loading head displacement rate was 10 mm/min. Based on the obtained load-deflection corves, the following bone strength parameters were determined: the maximum elastic strength (showing the relationship between force and deformation), the ultimate strength (equal to load leading to fracture and destruction of the bone structure), stiffness (as the slope of the load-deflection curve in of the region of elastic deformation), as well as elastic energy and work to fracture (as a area under load-deflection curve in elastic and total region of bone deformation). The analyses were performed using the Origin 2016 program (Origin Lab, Northampton, MA, USA) [22,24].

### 2.4. Geometrical Properties of the Tibia

Geometric properties of the tibia were determined based on measurements of vertical and horizontal diameters (external and internal) which were measured using a digital caliper. Assuming the elliptical shape of bone mid-shaft cross-sectional area, the following geometrical parameters of the tibia were calculated: mean relative wall thickness (MRWT), cortical cross-sectional area, cortical index, secondary moment of inertia, and radius of gyration [24].

### 2.5. Structural Properties of the Tibia

Structural parameters of the bones were calculated based on the measured strength characteristics and geometric parameters. These parameters describe the specific mechanical properties of the bone midshaft and are independent of bone size and the conditions under which the strength tests were performed. The following material properties were determined using Young’s modulus: elasticity, bending moment, toughness, yield strain, ultimate strain, yield stress, and ultimate stress [24].

### 2.6. Densitometry Measurements of the Tibia

To determine bone mineral density (BMD) and bone mineral content (BMC), dual-energy X-ray absorptiometry (DXA) measurements were made using a Discovery W densitometer (Hologic Inc., Bedford, MA, USA). BMD and BMC measurements were performed for the whole bone [22,25].

Bone tissue density (BTD) for a 5-cm-long mid-diaphyseal part was performed using an AccuPyc 1330 helium pycnometer (Micromeritics, Inc., Norcross, GA, USA). Before analysis, the samples were first defatted in 2:1 chloroform and methanol mixture for 12 h and then heated in an oven at 105 °C for 24 h to remove bound water and cooled at 25 °C in vacuum desiccators [26].

After assessing their densitometric properties, the bones were mineralized in a muffle furnace at 600 °C to determine the ash content. Ash content was expressed as a percentage of dry, defatted bone mass.

### 2.7. X-ray Diffraction

The crystallinity of bone mineral fraction was measured by X-ray diffraction (XRD). The resulting ash was ground in an agate mortar to form a fine white powder. The high-resolution X-ray Diffractometer (Empyrean, Panalytical, Almelo, The Netherlands) with Ni-filtered Cu K-alpha radiation (λ = 1.51874 Å) was used with a generator voltage of 40 kV and current of 30 mA. The samples were measured in Θ–2Θ geometry in the range of 10–100°, with a resolution of 0.01° and counts of 6 s per data point. Measurements were made at room temperature. The size of the hydroxyapatite crystallites in c-axis and a-b plane was determined based on the analysis of the shape of the peaks corresponding to the (002) and (310) Miller indices, respectively. The analyzes were performed using Origin software [27].

### 2.8. Composition of Bone Mineral Fraction

The composition of the mineral part of the bone tissue was determined in the ash obtained as a result of burning the midshaft part of the tibia. An ICP-OES spectrometer (iCAP Series 6500, Thermo Scientific, Waltham, MA, USA) was used to perform the analysis. Before spectrometry, the samples were mineralized using HNO_3_ and HCl. TraceCERT multielement stock solution (Sigma-Aldrich, St. Louis, MO, USA) was used as the reference standard. The content of macro- and microelements was determined in mg or µg per 1 g of dry, defatted bone mass [28].

### 2.9. Bone, Articular Cartilage, and Growth Plate Histomorphometry

Fragments of bone tissue containing trabecular bone, articular cartilage, and growth plate were cut from the proximal tibia using a diamond bandsaw (MBS 240/E, Proxxon GmbH, Foehren, Germany). From the midshaft part of the tibia, a fragment of the cortical bone was taken. The material was initially fixed in a 4% buffered formalin solution for 48 h, and then, the samples were demineralized in buffered EDTA (10%, pH 7.4). The decalcified samples were subjected to common histological procedures and finally embedded in paraffin. Five sections from each tibiae were cut from each animals with microtome (Microm HM 360, Microm, Walldorf, Germany). Sections were stained using the methods: Goldner trichrome, Safranin O, and Picrosirius red (PSR) [25,29]. Stained samples were analyzed using microscope (Olympus BX63; Olympus, Tokyo, Japan) in a brightfield (Goldner, Safranin O) or polarized light (PSR) [22].

Goldner trichrome staining was used to assess the morphology of the trabecular bone: the relative bone volume (BV/TV), trabecular thickness (Tb.Th), trabecular separation (Tb.Sp), and trabecular number (Tb.N). The obtained images were analyzed using the ImageJ (v.148, National Institute of Health, Bethesda, MD, USA) graphic analysis software [22].

Safranin O staining was used for analysis of basal histomorphometry of the articular cartilage, the growth plate, and the content of proteoglycans in the matrix. In the growth plate cartilage, the thickness of reserve, proliferation, hypertrophy, and calcification zones was measured. In the articular cartilage, the thickness of superficial, transitional, and deep zones was measured. Measurements were made at four locations along the cartilages. The obtained microscopic images were analyzed using Olympus cell-Sens (ver. 1.5, Olympus, Tokyo, Japan) graphics analysis software [22,25].

Picrosirus red (PSR) staining was used to assess the collagen structure in the trabecular and cortical bone. The identification and quantification of mature and immature collagen fibers in bone tissue was performed using ImageJ, which enabled automatic analysis based on differences in the color of immature (seen as green) and mature (seen as oragne-red) collagen fibers in polarized light [22,25]. The results were expressed as a percentage.

### 2.10. Statistical Analysis

Statistical analysis of the results was carried out using Statistica (v. 13.0, IBCO Software Inc., Palo Alto, CA, USA) software. All results are presented as mean values with standard deviations (mean ± S.D.). Before performing the significance tests, it was checked whether the examined features were normally distributed using the W. Shapiro–Wilk test, while the homogeneity of variance using the Levene test. Statistical significance was checked using Student’s *t*-test. If the examined features did not have homogeneous variances, the *t*-test with Welch’s correction was performed; non-parametric data were analyzed using a Mann–Whitney U test. The level of statistical significance was *p* < 0.05.

## 3. Results

There were no significant differences in body weight between the animals in studied groups.

### 3.1. Blood Serum Analysis

A significant decrease in IGF-1 (*p* < 0.01) in the experimental group was observed. The infection did not significantly affect insulin activity, GH, or calcitriol content. In addition, the level of bone turnover markers represented by osteocalcin and osteoprotegerin did not change significantly as a result of induced infection. In the case of biochemical indicators determined in the blood serum of animals infected with *Clostridium perfringens*, a significant decrease in the concentration of alkaline phosphatase (*p* < 0.01) and a significant increase in phosphorus (*p* < 0.05) and calcium (*p* < 0.001) was observed (Figure 1).

### 3.2. Bone Osteometry, Density, Geometry, Mechanical, and Structural Properties

Infection of *C. perfringens* between 17th and 20th day of age did not significantly affect the basic parameters of the tibia, such as length, weight, and length/weight bone ratio (Table 1). The densitometric analysis showed a significant decrease in bone density BMD (*p* < 0.05) in the animals from the experimental group (Table 1).

In the case of geometric parameters, the infection of animals with *Clostridium* contributed to a significant decrease in horizontal internal diameter (*p* < 0.05), cross-sectional area (*p* < 0.05), mean relative wall thickness (*p* < 0.05), and cortical index (*p* < 0.01). However, the value of the radius of inertia in the experimental group increased significantly (*p* < 0.001). The infection did not affect the remaining geometrical bone properties (Table 1).

A significant decrease in bone stiffness was observed in the experimental group compared to the control group (*p* < 0.01). Infection did not affect other mechanical properties (Table 2). For structural properties, statistical analysis showed a significant decrease in the Young’s modulus value (*p* < 0.001) and an increase in the yield strain (*p* < 0.01) in the experimental group. Other structural properties of the tibia did not differ significantly between both groups (Table 2).

### 3.3. Cancellous Bone Histomorphometry and Bone Collagen Structure

Histomorphometric analysis of the cancellous bone showed a statistically significant increase in the relative bone volume (BV/TVP < 0.001), number of trabeculae (Tb.N, *p* < 0.001), trabeculae thickness (Tb.Th, *p* < 0.001), and bone surface density (BS/BV, *p* < 0.001). As a result, a significant decrease of trabecular separation (Tb.Sp, *p* < 0.001) in the experimental group was observed (Figure 2).

Figure 3A,B presents the content of mature collagen fibers in the trabecular and cortical bone. Infection did not significantly affect the content of mature collagen fibers in the trabecular bone (Figure 3A,B). In cortical bone, *C. perfringens* infection significantly decreased the content of mature collagen fibers (*p* < 0.001) (Figure 3A,B).

### 3.4. Articular Cartilage and Growth Plate Cartilage Morphology

Histomorphometric analysis showed differences in the thickness of zones of growth plate cartilage, and a statistically significant increase in the thickness of the resting (*p* < 0.01) and hypertrophic zones (*p* < 0.001) was observed. On the contrary, the thickness of the proliferating (*p* < 0.001) and calcification zones (*p* < 0.01) was significantly reduced in the experimental group (Table 3, Figure 4).

In the case of articular cartilage, the superficial zone was significantly (*p* < 0.001) reduced in the infected group compared to the control group. In contrast, the transition zone was significantly (*p* < 0.001) thicker in infected animals. The thickness of the deep zone did not differ significantly between examined groups (Table 3, Figure 5).

### 3.5. Proteoglycans in Articular Cartilage and Growth Plate Cartilage

Staining with Safranin O showed an increasing gradient in the intensity of proteoglycan staining in individual zones of the growth plate cartilage, from the weakest in the resting zone to the strongest in the hypertrophic zone; however, the experimental group was characterized with the more intensive staining. Additionally, the proportion of proteoglycans in the proliferating zone was higher in infected animals. On the other hand, a higher content of proteoglycans in the resting zone was observed in the control group (Figure 4).

In the case of articular cartilage in animals from the experimental group, weaker intensity of proteoglycans staining in the matrix of the superficial zone and around the nutrient channels located in the transition zone was observed (Figure 5).

### 3.6. Ash Content and Composition of the Mineral Fraction in Bone

In animals from the experimental group, a significant reduction in the content of calcium (*p* < 0.05) and zinc (*p* < 0.01) in the mineral fraction of bone tissue was observed. However, in the case of copper (*p* < 0.01), statistical analysis showed a significant increase in this element in tibia taken from infected animals. The content of the other macro- and microelements did not differ significantly between examined groups (Table 4). 

### 3.7. Analysis of the Crystallinity of the Bone Mineral Phase

Animal infection also did not significantly affect the size of the hydroxyapatite crystals (Table 5).

## 4. Discussion

Proper nutrition is one of the factors affecting the bone tissue remodeling process, including supply of minerals and calcium especially, as it is the inherent building block of bone tissue. Phosphorus is also very important as well as maintaining a balance between the amount of Ca and P in the body. However, not only does the right amount of nutrients affect the maintenance of normal BMD but also the proper absorption of these elements in the digestive tract [30]. This process is influenced by microorganisms inhabiting the gastrointestinal tract, which influence the proper absorption of all minerals regardless of the type of food consumed [30,31].

In recent years, studies have been conducted that confirm that changes in the intestinal microbiome affect the mineral density of bone tissue and, consequently, increase the risk of the development of skeletal diseases (osteoporosis and inflammatory joint diseases) [32]. Disturbances in the composition of the microflora have been associated with inflammatory bowel diseases (IBD) [33]. Clinical studies show that in IBD patients, the composition of the microflora is different compared to healthy patients [34]. Numerous studies have also shown that the administration of probiotics, antibiotics, and the transplantation of fecal microbiota, which change the composition of the microflora in people suffering from IBD, significantly alleviate the symptoms. As well as in the available literature, it can be found that pathogenic microorganisms can exacerbate the course of IBD [35,36]. One of the pathological bacteria influencing the development of IBD may be *C. perfringens*. Study conducted in children and adolescents with IBD proved the presence of this bacterium in 9% of cases [36].

Common inflammatory bowel diseases (IBD) include Crohn’s disease and ulcerative colitis, which belong to the group of chronic inflammatory diseases of the gastrointestinal tract. The incidence of osteoporosis in IBD patients is 5–40% and 16–77%, respectively [37]. Moreover, the risk of low-energy fractures is much higher and reaches to 40–60%. Although osteoporosis occurs more often in females, the risk of developing this bone disease in both genders is equal in IBD patients [37]. IBD also occurs in young people and children. In children suffering from IBD, stunting is observed, and growth deficits may become permanent. In diagnosed, untreated children, a decrease in the rate of bone turnover in the trabecular bone and a decrease in the content of bone turnover markers are observed [38].

There is no information in the available literature about the effect of gas gangrene caused by the bacterium *C. perfringens* on the process of modeling and remodeling bone tissue. Disruptions in the microenvironment of the bacterial flora can lead to the pathological multiplication of this gram-positive bacteria. Research over the past few years has shown that infection with *C. perfringens* causes a strong inflammatory reaction in the intestines. The intestinal mucosa becomes hyperemic and is covered with many inflammatory cells [39]. The study of Park et al. (2008) showed an increase in the levels of interleukin (IL)-1β, IL-10, interferon (IFN)-α, and INF-γ in lymphocytes collected from the intestinal endothelium of *C. perfringens*-infected animals [40]. This study also showed an increase in the levels of IL-4, IL-10, and IFN-γ in animals from the infected group [41]. The immune reaction is caused not only by the bacterium itself but also by the toxins (mainly the α-toxin) that are produced by it. The α-toxin in gas gangrene in humans increases the platelet count at sites of infection, which will compromise the function of neutrophilia. Studies conducted on umbilical vein endothelial cells also showed that the α-toxin contributes to the increase in the level of IL-8 and tumor necrosis factor alpha (TNF-α) in the leukocytes of these cells [39,42]. In an animal study performed by Shuang et al. (2015), when the bacterium and the toxin were administered simultaneously, an increase in the expression of IL-6 IL-8, iNOS (inducible nitric oxide synthase), and TNF-α mRNA was demonstrated [39].

Inflammation of the intestinal mucosa contributes to the impairment of nutrient absorption, including minerals. In addition, the inflammation itself causes pain, leading to the reduction of food intake [43]. Adequately, the supply of minerals and their proper absorption have a significant impact on the processes of bone tissue remodeling and the maintenance of normal BMD [30]. Studies have also shown that pro-inflammatory cytokines, such as TNF-α, IL-1β, IL-6, and INF-γ, have an influence on maintaining the correct balance between osteogenic and osteolysis processes. These cytokines affect, among others, OPG, which significantly affects the processes taking place in the bone tissue [44]. Research has also shown that genetic variations in the IL-6 affect bone loss in patients with IBD [45].

Diseases of the gastrointestinal tract caused by bacteria of the genus *Clostridium* often appear in the elderly and during hospitalization, which may intensify the degenerative processes of bone tissue [18,46].

In the presented experiment, chicken broilers were selected as the model animal. These animals are characterized by a rapid growth rate compared to other models. These birds reach somatic maturity at the age of 42 days, and their high daily gains make possible to induce noticeable changes in the metabolism of bone tissue in a short time; thus, broiler chickens are year by year more frequently used in experiments in which the skeletal system is studied [20,21]. The studies conducted over the last few years show that broiler chickens are a very good model in preclinical studies assessing organism growth. Additionally, this bird model is used in experiments assessing malnutrition in organisms during development. All these indicate that broiler chickens can be used to study the effect of disturbances in the gastrointestinal microflora, which in turn significantly affect the absorption of nutrients [20]. Numerous recent studies showed changes in gene expression of proteins are responsible for intestinal tight junction integrity or barrier function (claudin-1, occludin, mucin-2, Fatty acid-binding protein-2) in broiler chickens during gut inflammation induced by *C. perfringens* challenge, which are similar to those observed in humans during the IBD [47,48,49,50]. Broiler chickens are also sensitive to *C. perfringens*, which facilitated the infection process itself. Moreover, the tibia is considered to be the model bone for such studies [20,51].

In the presented experiment, it was found that the infection with the gram-positive *Clostridium perfringens* bacterium had no effect on the weight and length of the examined bone. The lack of changes in these parameters is most likely caused by the short-term exposition period and the young age of the experimental animals, which could reduce the negative effect of the tested factor.

Densitometric analysis showed a negative effect of *C. perfringens* infection on the bone tissue mineralization process. In the experimental group, the mineral density decreased by 8%. Young birds were infected, and a decrease in BMD may contribute to the formation of abnormal peak bone mass and thus increase the risk of osteoporosis later in life [52]. Probably, this bacterium causes inflammation of the intestinal mucosa, which contributes to the malabsorption of nutrients, including minerals [43], leading to the reduction of BMD [30]. This bacteria and the toxins produced by it increase the level of pro-inflammatory cytokines, including TNF-α, IL-1β, IL-6, and INF-γ [39,40,41]. It has also been shown that these cytokines contribute to the maintenance of the correct balance between osteogenic and osteolysis processes through their influence on bone turnover markers, including OPG [44].

In our work, the level of OPG did not differ significantly between the studied groups, which may be related to the triggering of not-too-severe infection. However, changes in BMD were still observed. These may indicate a further imbalance between the processes of bone mineralization taking place in the bone tissue. In the research of Sjögren et al. (2012), the increase in BMD was observed, like in the research of Schwarzer team (2016). In both studies, no differences in mineral density were found. Such results are fully contrasting to our findings, where a significant decrease in BMD was observed. The different results may be due to the fact that in the experiments of previously mentioned researchers, the bacterial flora of the gastrointestinal tract were completely removed, whereas in the presented experiment, only the composition of the microflora was disturbed by the administration of C. *perfringens* for three days. Moreover, both of the experiments of Sjögren et al. (2012) and Schwarzer et al. (2016) were carried out on newborn animals, while in our study, the microflora composition was disturbed during the growth of the animals [53,54]. Such a dissimilarity makes a crucial difference. Similarly to our finding, the decrease in BMD was obtained in the experiment carried out by Irwin et al. (2013), which investigated the effect of *Helicobacter hepaticus*-induced ulcerative colitis on bone metabolism on already fully developed skeleton of 14-week-old mice [55]. The reasons of differences should therefore be justified by the age of the model organisms or by the method of inducing changes in the microflora of the gastrointestinal tract. [53,54,55].

The presented results show that a short-term exposure to the bacterium *C.* affects the mineral density of bone tissue. Changes in BMD of the tested animals due to infection with *C. perfringens* reflect changes in the composition of the mineral fraction of bone tissue. The infection significantly decreased the content of zinc and calcium by 5% and 1.5%, respectively. However, in the case of copper, a significant increase of 6% in the experimental group was observed. 

Calcium ions are the basic building block of bone tissue and are responsible for maintaining proper mineral balance in the body [56]. Copper and zinc are their antagonists. In the case of an increase in copper content, a decrease in zinc content can be observed, which was observed in the results obtained in the presented study. Zinc is a very important element for the body, including the skeletal system, because it affects the process of bone growth and development by reducing the activity of osteoclasts. Long-term gastrointestinal diseases affect the zinc content [56]. On the other hand, copper influences the synthesis of collagen fibers. Copper deficiency may weaken the collagen structure of bone tissue and also has a negative effect on bone formation, increasing the risk of skeletal diseases [57].

In the presented study, differences in the content of calcium and phosphorus in the blood serum were also observed. In animals from the experimental group, the infection contributed to an increase in the content of these macronutrients by 37% and 7%, respectively. The alkaline phosphatase also decreased by 224% due to infection with the bacterium of the genus *Clostridium*. Alkaline phosphatase is responsible for binding calcium and phosphorus in the bones. The decrease in its activity contributes to a reduction in the amount of Ca and P deposited in the bone tissue, increasing their concentration in the blood serum [58].

Despite such clear connections, there is little information in the available literature on the impact of the changes in the homeostasis of the gastrointestinal microenvironment to strength properties of bone tissue [24]. In the presented study, the analysis of strength parameters only shows that bone stiffness has been changed. In the infected group, this parameter was reduced by 21% compared to the control group.

Bone stiffness is largely determined by its organic phase, consisting mainly of collagen. The other determined bone tissue strength parameters did not change, which is a consequence of the changes in the shaft geometry that took place in the experimental group. Lower BMD was observed in this group. In order for the bone to perform its supporting functions related to maintaining a rapidly growing organism, it must be subject to adaptive changes in geometry. These changes have been observed in our study. Infection with *C. perfringens* contributed to the reduction of the following geometrical parameters of the stem: cross-sectional area, mean relative wall thickness of the body of the cortical index by 10%, 12%, and 11%, respectively. The infections also contributed to an increase in the inner horizontal diameter of the tibia shaft and the radius of gyration by 11% and 7%, respectively. The increase in bone diameter could have contributed to changes in the cross-sectional area in the experimental group [24].

It is known that maximum elastic strength or ultimate strength depend on bone mineralization, its shape, and length [24]. Adaptive changes occurring in the geometry of less mineralized bone may mean that bone strength, defined as the value of the breaking force, may not change [27], which was also observed in the presented experiment. Therefore, more complete information on changes in the structure of bone tissue can be obtained by analyzing changes in material parameters, which do not depend on the shape and size of the bone. In the case of structural parameters, an increase in yield strain and a decrease in the Young’s modulus by 38% and 27%, respectively, were observed. Changes in these properties of bone tissue materials again indicate changes in the collagen structure of the bone cortical part, which was reflected in the decrease in bone stiffness. In the experimental group, the content of immature collagen fibers increased and the ration of mature to immature collagen was reduced by 44%. The greater content of immature weakly mineralized collagen fibers contributed to a reduction in stiffness and the related increase yield strain and a reduction in the value of Young’s modulus of less mineralized bone with lower BMD [27]. Changes in the content and the ration of mature to immature collagen may be related to the increased of copper content in the mineral fraction because Cu affects the synthesis of collagen fibers [57].

In 2017, Guss et al. conducted an experiment in which they examined the effect of disturbances in the intestinal microflora on the mechanical and geometric properties of bone tissue. For this purpose, two animal models were used: TLR5KO mice lacking the toll-like 5 receptor and wild-type C57BL/6J mice [59]. The absence of the toll-like 5 receptor disrupts the host-microflora relationship. In order to disturb the intestinal microflora, broad-spectrum antibiotics (ampicillin, neomycin) were used between 4th and 16th week of age (from weaning to skeletal maturity) [59]. Long-term use of antibiotics to induce changes in the intestinal microflora resulted in a decrease in cortical bone density and bending moment values in TLR5KO mice by 21% and C57BL/6J by 9% [59]. Changes in geometric parameters were also observed in this experiment. In TLR5KO individuals, it was found that the altered intestinal microflora contributed to a decrease in the total bone marrow and cortical surface. On the other hand, C57BL/6J mice showed disturbances in the microflora of the gastrointestinal tract that contributed to an increase in the marrow area and a reduction in the cortical area [59] although the range of changes in the mechanical properties of bone obtained by Guss et al. (2017) are different from those presented in this work [59]. This study also clearly shows the negative impact of disturbances in the intestinal flora on the strength and geometric properties of bones.

Changes in bone geometric properties were also observed in patients with IBD, where the effects of Crohn’s disease and ulcerative colitis on the geometric properties of hand bones were compared. Crohn’s disease has been shown to cause much greater changes in cortical bone geometry than ulcerative colitis [60].

Infection with a bacterium of the genus *Clostridium* also influences the histomorphometry of the trabecular bone, contributing to a significant increase in the volume of bone trabeculae as a result of an increase in their number and thickness by 36%, 22%, and 17%, respectively. In addition, the bone surface density increased by 16%. These changes reduced the trabecular separation 36%.

In the experiment of Sjögren et al. (2012), they found that the lack of intestinal microflora in newborn females contributed to the weakening of osteoclast processes while not inhibiting bone formation [53]. In the group devoid of intestinal microflora, an increase in BV/TV and the number of bone trabeculae was also observed, with no changes in their thickness at the seventh and ninth week of life. However, there were no changes in the rate of trabecular bone mineralization [53]. 

Similar changes in the trabecular bone are observed in *C. perfringens* infection, which could have impaired the relationship in osteolytic and osteogenetic processes while affecting the mineralization process. In our experiment, densitometric analysis showed a decrease in BMD for entire tibia in infected animals. Additionally, in the experimental group, a decrease in the activity of alkaline phosphatase and changes in the content of calcium and phosphorus in the blood were observed with the simultaneous unchanged concentration of the examined markers of bone turnover in the blood serum. The decrease in the activity of alkaline phosphatase informs about an impaired process of calcium and phosphorus binding in the bone [58]. It should also be noted that in the tubercular bone, as in the case of cortical bone, the content of maturing collagen increased, which may also confirm the weakening of osteolytic processes in the absence of inhibition of osteogenetic processes. The observed results may suggest that the trabecular bone will be characterized by lower strength. Later in life, the risk of the development of osteoporosis and the appearance of low-energy fractures can increase.

In children with IBD, it was found that as a result of treatment the level of bone turnover markers returns to normal, with a continuous lack of adequate bone mineralization [38]. Changes in the trabecular bone are also observed in the group where ulcerative colitis was caused. In the males, a decrease in the volume of bone tissue in the tubercular bone was found, while in the females, no similar changes were observed [55]. Changes in the histomorphometry of the trabecular bone are also observed in patients with Crohn’s disease [61].

The results presented in this paper and those obtained by Sjögren et al. [53] in the case of the histomorphometric parameters of the trabecular bone differ from the results obtained by Irwin et al. [55] or in patients with IBD, where the histomorphometric parameters of the bone trabeculae decreased. These differences may be due to the different ages of the studied individuals. Moreover, Oostlander et al. [61] showed that the gender of the studied individuals influences the observed changes. The presented study and experiment by Sjögren et al. [53] was carried out on young, growing animals. On the other hand, the experiment by Irwin et al. [55] was conducted on mice at the age of 14 weeks, which were characterized by skeletal maturity. Therefore, it can be concluded that the lack of gastrointestinal microflora or the disturbance of their composition affects the histomorphometry of trabecular bone depends on age.

Infection with the bacterium *C. perfringens* also affects the histomorphometry of the growth plate and articular cartilage. In the case of articular cartilage, infection contributed to a 29% reduction in the thickness of the superficial zone and an increase in the thickness of the transition zone by 14% compared to the control group. The changes that we observe as a result of disturbed intestinal microflora homeostasis in this study may affect the function of articular cartilage. Zone I thickness reduction can be considered as a negative effect of *C. perfringens* infection on the structure of articular cartilage [62,63].

The lower expression of proteoglycans in the superficial zone of articular cartilage and around the nutrient channels indicates that *C. perfringens* infection also affects the metabolic activity of cartilage cells (Figure 4 and Figure 5). Toxins produced by bacteria of the genus *Clostridium* may decrease the content of proteoglycans by impaired water absorption in the gastrointestinal tract and disturbances in the electrolyte balance, including the content of sodium ions, which affect the ability of good cells to absorb water [62]. It is thanks to this ability that chondrocytes are responsible for the elasticity of cartilage. Even if these changes occur in young animals, they can contribute to the development of inflammatory joint diseases later in life.

Szychlinska et al. [64] presented the impact of disturbances in the intestinal microflora linked with the appearance of degenerative changes in articular cartilage. It was suggested that the microorganisms inhabiting the digestive tract affect the entire physiology of the body, and its composition may change under the influence of various factors, which may contribute to the appearance of degenerative joint diseases [64]. To date, there are no available experimental results confirming this hypothesis.

In this study, the changes caused by the infection also concern the growth plate cartilage. Histomorphometric analyses showed that the resting and hypertrophic zones thickness is reduced by 22% and 60%, respectively. On the other hand, the proliferation and calcification zones were reduced in the experimental group by 15% and 7% in comparison to the control group. Changes in the individual zones that make up the growth plate cartilage may contribute to the impairment of its function, which is responsible for bone growth through cartilage ossification [65]. A very recent study shows the relationship between the number of *Streptococcus spp.* and the intensity of pain perception in patients suffering from osteoarthritis, but there is no explanation of the mechanisms connected with these two factors [66]. On the other hand, research conducted on rats with induced arthrosis shows that the use of moxibustion affects the articular cartilage through the microflora of the gastrointestinal tract. However, there is no more detailed information about the two mechanisms [67].

The experiment carried out on adult mice lacking the bacterial microflora showed that these animals were characterized by a smaller thickness of the growth cartilage and a lower metabolic activity of cartilage cells. This means that microorganisms inhabiting the gastrointestinal tract affect the growth plate and thus bone growth along the length [68].

Each zone that makes up the growth plate cartilage plays a different role. The functions of the resting zone are not fully understood. In an experiment by Abad et al. [69], the resting zone thickened as a result of surgical removal of the proliferating and hypertrophic zones. The changes in the thickness of the resting and proliferating zones observed in the present study compared to the work of Abad et al. [69] may indicate that as the proliferating zone was reduced, the resting zone is thickened to support the proliferating zone in its functions. However, not only do changes in zone thickness negatively affect the functions of cartilage but also changes in the metabolic activity of chondrocytes, which can be expressed in the amount of proteoglycans in the matrix [69]. In the cartilages of experimental animals, the content of proteoglycans in the resting zone is lower than in the control group, while in the proliferating and hypertrophic zones, the metabolic activity of the cartilage cells is higher compared to the control group (Figure 4 and Figure 5). In the proliferating zone, this may be due to the fact that the zone itself has been reduced, and the cartilage cells tend to maintain homeostasis. In the hypertrophic zone, we also observe a greater amount of proteoglycans with a simultaneous thickening of the zone, which may be related to the fact that, compared to the control group, it contains more metabolically active cells. The presented experiment was carried out on young birds. Disturbances in the thickness of individual zones and the activity of proteoglycans could have contributed to changes in the histomorphometry of the trabecular bone, which, in the long term, may lead to the appearance of skeletal system diseases.

It has been shown that the microflora influences the endocrine system by influencing the metabolic processes occurring throughout the body. When mice were treated with antibiotics to disturb the composition of the gastrointestinal tract, a decrease in the amount of IGF-1 in the blood serum was observed. [33,68]. Similarly, in our study, a 46% decrease in IGF-1 activity was observed in animals from the experimental group. IGF-1 affects the synthesis of proteins, including proteoglycans and collagen; it also influences the differentiation and growth of chondrocytes and the amount of sulphates in the cartilage tissue matrix, which, together with hyaluronic acid, form proteoglycan aggregates. IGF-1 also affects the degree of bone tissue mineralization [54]. 

## 5. Conclusions

The presented work is the first one showing that even short-term exposure to *C. perfringens* can affect bone metabolism. The presented changes in mineral density are small, and there are also no changes in the osteometric parameters of the tibia, which may be related to the young age of the studied animals and the short period of infection. Nevertheless, observed changes in the growth plate and the articular cartilage histomorphometry could lead to bone and cartilage diseases in later life.

## Figures and Tables

**Figure 1 jcm-11-00205-f001:**
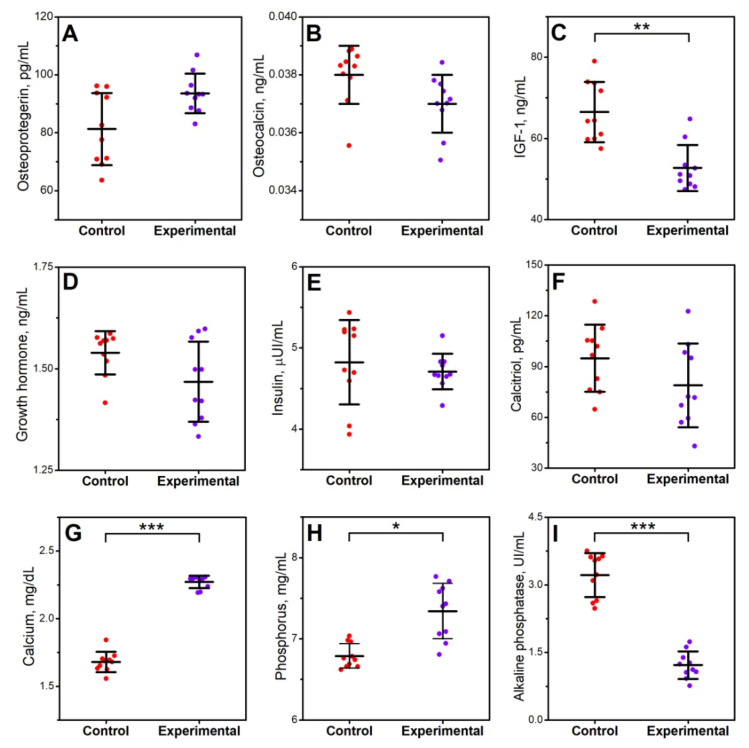
The effect of *C. perfringens* infection on hormonal and biochemical analysis in blood serum of animals at the age of 42 days: (**A**) osteprotegerin; (**B**) osteocalcin; (**C**) insulin-like growth factor 1 (IFG-1); (**D**) growth hormone; (**E**) insulin; (**F**) calcitriol; (**G**) calcium; (**H**) phosphorus; and (**I**) alkaline phosphatase. Control—the control group of birds; experimental—the experimental group of birds subjected to *Clostridium perfringens* infection. Data are mean values ± SD (whiskers) from *n* = 10 birds. Statistical significance: * *p* < 0.05; ** *p* < 0.01; *** *p* < 0.001 (two-tailed Student’s *t*-test, *t*-test with Welch’s correction, or Mann–Whitney U test).

**Figure 2 jcm-11-00205-f002:**
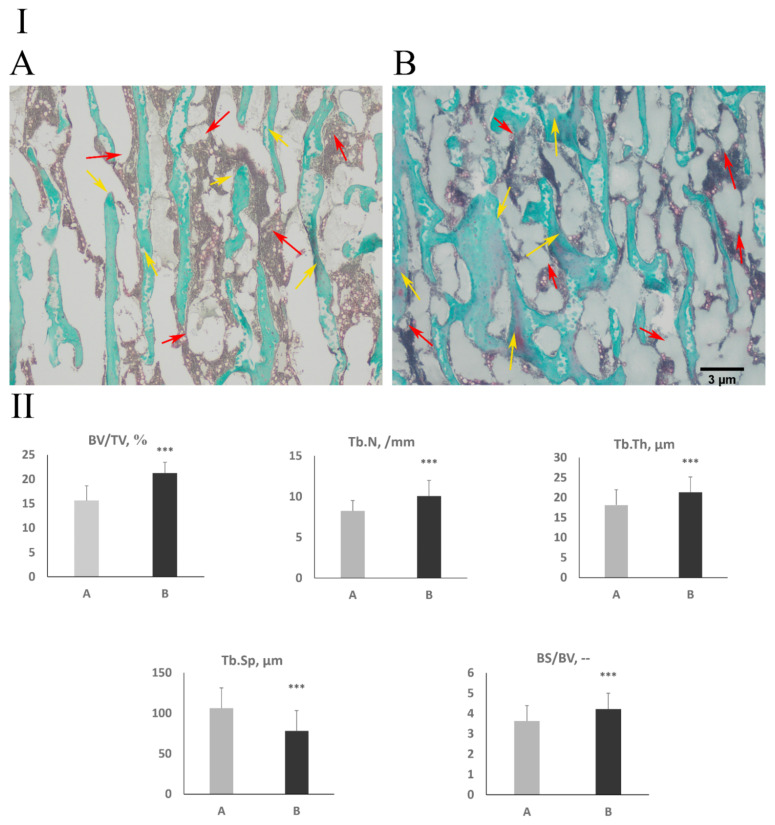
Trabecular bone morphology from proximal part of the tibia bone in examined animals of the age of 42 days. (**I**) Representative pictures of Goldner Trichome method staining carried out on formaldehyde-fixed samples from the cancellous bone. The yellow arrows show the trabeculae, while the red arrows show the bone marrow. (**II**) BV/TV-relative bone volume. Tb.Th, trabecular thickness; Tb.Sp, trabecular separation; Tb.N, trabecular number; BTD, bone tissue density; BS/BV, bone surface density. Control group (A); experimental group (B). Data are mean values ± SD (whiskers) from *n* = 10 birds. Statistical significance: *** *p* < 0.001 (two-tailed Student’s *t-*test, *t*-test with Welch’s correction, or Mann–Whitney U test).

**Figure 3 jcm-11-00205-f003:**
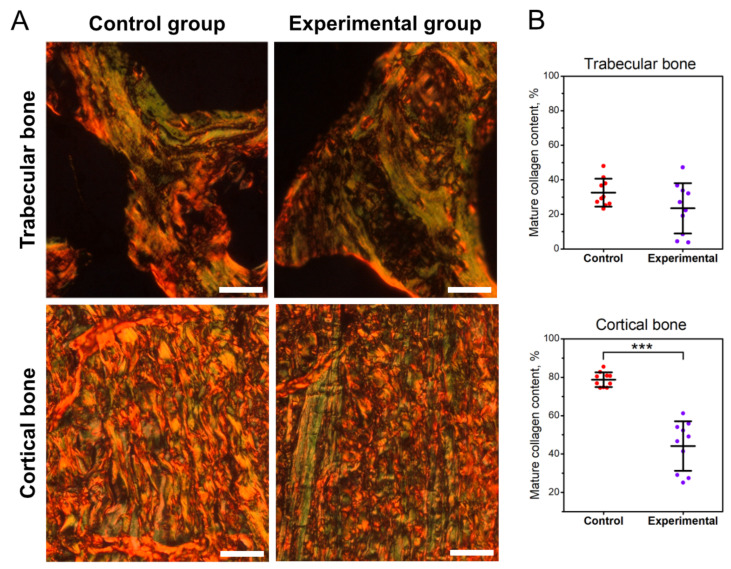
The effect of *C. perfringens* infection on the bone content of mature collagen in tibia of animals at the age of 42 days: (**A**) representative photos of the PSR staining carried out on formaldehyde-fixed from the trabecular (upper panels) and cortical bone (lower panels) of: mature, thick collagen, which is red or orange, and immature, thin collagen, which is green. All the scale bars represent 10 µm. (**B**) Content of mature, thick collagen fibers in the trabecular (upper graph) and cortical bone (lower graph) of tibia. Control, the control group of birds; experimental, the experimental group of birds subjected to *C. perfringens* infection. Data are mean values ± SD (whiskers) from *n* = 10 birds. Statistical significance: *** *p* < 0.001 (two-tailed Student’s *t*-test, *t*-test with Welch’s correction, or Mann–Whitney U test).

**Figure 4 jcm-11-00205-f004:**
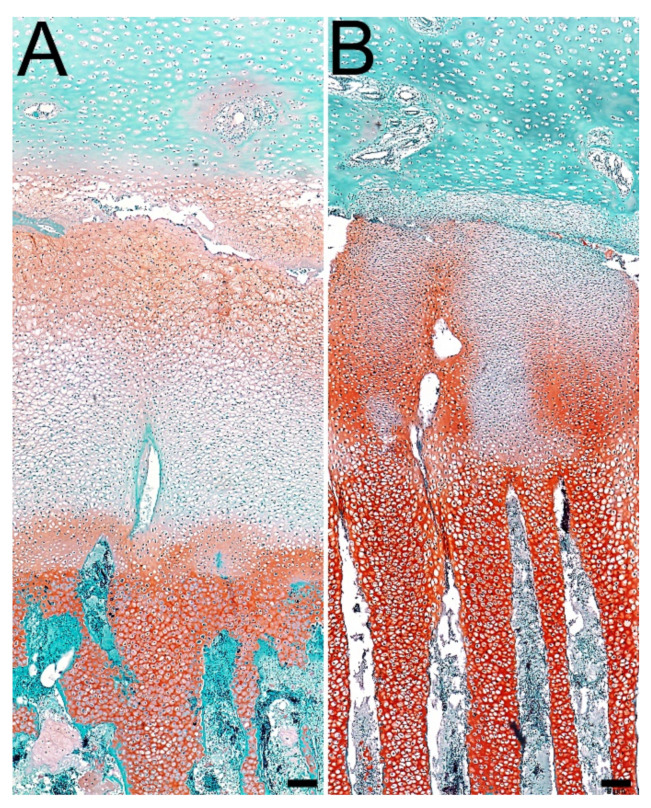
Representative photos of Safranine O staining carried out on formaldehyde-fixed sections from the growth plate cartilage of tibia of animals at 42 days of age. Control group (**A**); experimental group (**B**). Differences were observed in proteoglycan staining in tibial growth plate cartilage. Staining with Safranin O indicates the presence of proteoglycans, the red color in the matrix. All the scale bars represent 8 µm.

**Figure 5 jcm-11-00205-f005:**
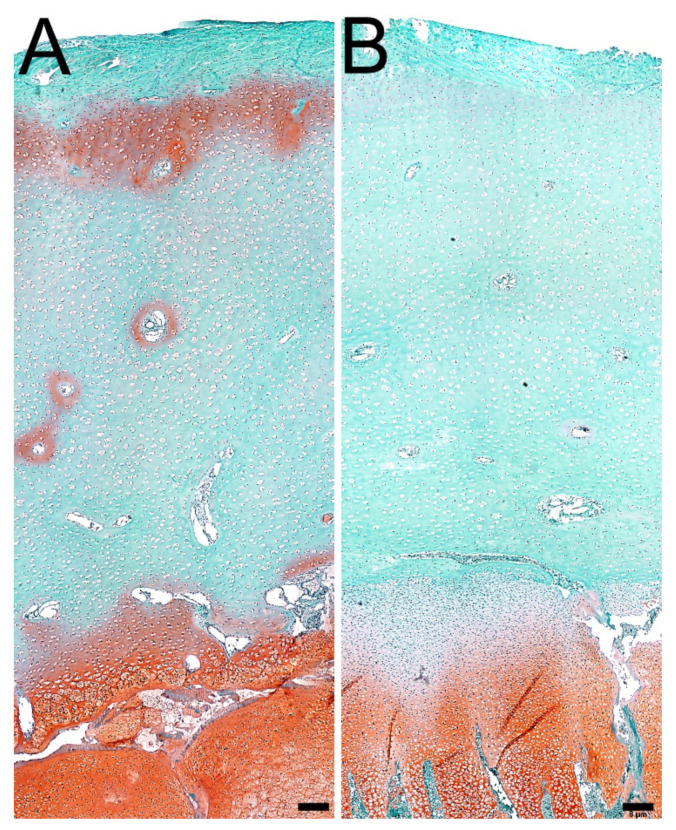
Representative photos of Safranine O staining carried out on formaldehyde-fixed sections from the articular cartilage of tibia of animals at 42 days of age. Control group (**A**); experimental group (**B**). Differences were observed in proteoglycan staining in tibial articular cartilage. Staining with Safranin O indicates the presence of proteoglycans, the red color in the matrix. All the scale bars represent 8 µm.

**Table 1 jcm-11-00205-t001:** The effect of *C.*
*perfringens* infection on osteometry, densitometry, and geometrical properties of the tibia in examined animals at the age of 42 days.

	Control	Experimental	*p*-Value
Bone osteometric properties
Bone weight, g	20.8 ± 1.55	21.7 ± 1.82	NS
Bone length, mm	1131 ± 36.0	1152 ± 29.7	NS
Length/weight bone ration	18.4 ± 1.17	18.8 ± 1.33	NS
Bone densitometry
BMD, g/cm^2^	0.267 ± 0.015	0.248 ± 0.012	*
BMC, g	3.52 ± 0.326	3.51 ± 0.228	NS
BTD cortical bone, g/cm^3^	1.87 ± 0.047	1.91 ± 0.086	NS
Bone geometrical properties
Horizontal external diameter, mm	9.76 ± 0.399	9.94 ± 0.658	NS
Horizontal internal diameter, mm	5.81 ± 0.506	6.44 ± 0.592	*
Vertical external diameter, mm	7.73 ± 0.460	7.98 ± 0.522	NS
Vertical internal diameter, mm	5.09 ± 0.451	5.44 ± 0.499	NS
Cortical cross-sectional area, mm^2^	36.6 ± 2.606	33.4 ± 3.24	*
MRTW	0.61 ± 0.082	0.51 ± 0.083	*
Cortical index, %	37.7 ± 2.78	33.0 ± 2.64	**
Secondary moment of inertia, mm^4^	192 ± 28.54	197 ± 29.7	NS
Radius of gyration, mm	2.24 ± 0.056	2.40 ± 0.090	***

Data given are mean (*n* = 10) with corresponding standard deviations; *p-*values; NS, not significant; * *p* < 0.05; ** *p* < 0.01; *** *p* < 0.001. BMD, bone mineral density; BMC, bone mineral content; BTD, bone tissue density; MRWT, mean relative wall thickness.

**Table 2 jcm-11-00205-t002:** The effect of *C. perfringens* infection on biomechanical properties of the tibia in examined animals at the age of 42 days.

	Control	Experimental	*p*-Value
Mechanical properties			
Max. elastic strength, N	211 ± 23.7	215 ± 20.2	NS
Elastic energy, MJ	64.5 ± 10.6	77.8 ± 11.4	NS
Ultimate strength, N	378 ± 35.6	389 ± 50.6	NS
Work to fracture, MJ	509 ± 59.0	520 ± 32.1	NS
Stiffness, N/mm	375 ± 8.07	310 ± 7.86	***
Structural properties			
Young’s modulus of elasticity, GPa	3867 ± 160	3037 ± 224	***
Bending moment, Nm	23.4 ± 1.91	23.6 ± 1.66	NS
Toughness Modulus, mJ/mm^3^	0.268 ± 0.057	0.279 ± 0.063	NS
Yield strain	0.013 ± 0.001	0.018 ± 0.003	**
Yield stress, MPa	520 ± 63.0	471 ± 53.4	NS
Ultimate strain	0.045 ± 0.002	0.047 ± 0.006	NS
Ultimate stress, MPa	836 ± 97.1	892 ± 66.3	NS

Data given are mean (*n* = 10) with corresponding standard deviations; *p-*values; NS, not significant; ** *p* < 0.01; *** *p* < 0.001.

**Table 3 jcm-11-00205-t003:** The effect of *C. perfringens* infection on growth plate and articular cartilage histomorphology of the tibia in examined animals at the age of 42 days.

	Control	Experimental	*p*-Value
Growth plate cartilage			
Resting zone, µm	311 ± 71.0	379 ± 89.2	**
Proliferation zone, µm	1001 ± 176	864 ± 213	***
Hypertrophy zone, µm	812 ± 132	1297 ± 135	***
Calcification zone, µm	1295 ± 101	1207 ± 199	**
Articular cartilage			
Surperficial zone, µm	85.1 ± 10.3	65.8 ± 11.0	***
Transition zone, µm	613 ± 52.0	699 ± 20.8	***
Deep zone, µm	177 ± 19.9	181 ± 15.2	NS

Data given are mean (*n* = 10) with corresponding standard deviations; *p-*values; NS, not significant; ** *p* < 0.01; *** *p* < 0.001.

**Table 4 jcm-11-00205-t004:** The effect of infecting *C. perfringens* on the content of ash, macro-, and microelements in the bone tissue of the tibia in examined animals at the age of 42 days.

	Control	Experimental	*p*-Value
Ash, %	58.8 ± 2.18	56.9 ± 3.18	NS
Ca, g/kg	494 ± 3.68	487 ± 10.0	*
P, g/kg	258 ± 3.16	259 ± 2.98	NS
Mg, g/kg	10.5 ± 0.175	9.91 ± 0.225	NS
S, g/kg	11.4 ± 0.239	11.5 ± 0.135	NS
Na, g/kg	3.18 ± 0.156	3.16 ± 0.140	NS
Co, mg/kg	0.351 ± 0.039	0.347 ± 0.033	NS
Cu, mg/kg	27.4 ± 0.702	29.1 ± 1.18	**
Fe, mg/kg	294 ± 17.3	286 ± 21.58	NS
Se, mg/kg	1.21 ± 0.109	1.09 ± 0.168	NS
Si, mg/kg	4.05 ± 0.744	3.14 ± 1.46	NS
Zn, mg/kg	445 ± 14.8	423 ± 13.5	**

Data given are mean (*n* = 10) with corresponding standard deviations; *p-*values; NS, not significant; * *p* < 0.05; ** *p* < 0.01.

**Table 5 jcm-11-00205-t005:** The effect of infecting *C. perfringens* on the hydroxyapatite nanocrystallites size in the bone tissue of the tibia in examined animals at the age of 42 days.

	Control	Experimental	*p*-Value
a-b plane, µm	69.0 ± 2.23	68.0 ± 2.32	NS
c axis, µm	94.5 ± 1.73	95.8 ± 1.19	NS

Data given are mean (*n* = 10) with corresponding standard deviations; *p*-values; NS, not significant.

## Data Availability

The data presented in this study are available on request from the corresponding author.

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
