# Peer review of "Structural Changes in Trabecular Bone, Cortical Bone and Hyaline Cartilage as Well as Disturbances in Bone Metabolism and Mineralization in an Animal Model of Secondary Osteoporosis in Clostridium perfringens Infection"

_jcm, 2021, doi:10.3390/jcm11010205_

Round 1

Reviewer 1 Report

Please see my additional comments in Red. 

Reviewer 2 Report

The study under review describes the effects of Clostridium perfringens infections on trabecular bone, cortical bone and hyaline cartilage of the growing chick. A large number of histomorphometric and serological parameters were measured to describe the influence of this intestinal infection on bone metabolism and mineralization.

The study is well planned and constructed and presents an exhaustive array of related results. The animal model chosen is suited for the clarification of the problem under study. Material and methods are sufficiently and reproducibly described. The manuscript is well structured well.

The study subject is of relevance to the clinician as well as to those interested in basic research.

The weakest point of the manuscript is the somewhat inexpert use of the English language. Some editing by a native English speaker seams highly recommendable.

For instance:

  • The calcification zone of the growth plate is called calcium zone
  • Line 381 “shows” instead of “shown”
  • Line 385 “study groups” instead of “studies groups”
  • Line 428 “factors” instead of “factor”
  • Line 442 “show” instead of “shows”
  • Line 448 “bacteria”? instead of “batteries”
  • And many more

Some sentences are incomplete and their meaning is hard to discern. For instance, Lines 336-342.

As an option the manuscript might win if the results are compared to diseases like rickets which shows similarities in some parameters like enlarged calcification zone of the growth plate and thicker trabeculae because of a higher amount of osteoid with delayed mineralization.
